# Golgi stress induces SIRT2 to counteract *Shigella* infection via defatty-acylation

Miao Wang[1,3], Yugang Zhang[1,3], Garrison P. Komaniecki[1,3], Xuan Lu[1], Ji Cao[1], Mingming Zhang[1,2], Tao Yu[1], Dan Hou[1], Nicole A. Spiegelman[1], Ming Yang[1], Ian R. Price ®[1] & Hening Lin ®[1,2] ✉

Enzymes from pathogens often modulate host protein post-translational modifications (PTMs), facilitating survival and proliferation of pathogens. *Shigella* virulence factors IpaJ and IcsB induce proteolytic cleavage and lysine fatty acylation on host proteins, which cause Golgi stress and suppress innate immunity, respectively. However, it is unknown whether host enzymes could reverse such modifications introduced by pathogens' virulence factors to suppress pathogenesis. Herein, we report that SIRT2, a potent lysine defatty-acylase, is upregulated by the transcription factor CREB3 under Golgi stress induced by *Shigella* infection. SIRT2 in turn removes the lysine fatty acylation introduced by *Shigella* virulence factor IcsB to enhance host innate immunity. SIRT2 knockout mice are more susceptible to *Shigella* infection than wildtype mice, demonstrating the importance of SIRT2 to counteract *Shigella* infection.

Protein post-translational modifications (PTMs) are essential for normal physiology as well as pathogenesis. In the past decades, emerging evidence have demonstrated that pathogens utilize various PTMs as a key strategy to modulate host proteins for their optimal survival and proliferation[1,2]. Such PTMs, including ADP-ribosylation[3], adenylation or AMPylation[4–6], and ubiquitination[7], target important pathways of host cells, including translation[8], signaling[9], membrane trafficking[10], and cytoskeleton arrangement[11,12]. Recently, it was reported that *Shigella flexneri* also modulates different host protein PTMs. *S. flexneri* effector, IpaJ, induces Golgi fragmentation through the cleavage of N-myristoylated host proteins[13]. The Golgi apparatus, apart from its role in membrane trafficking, also serves to integrate and transduce stress stimuli. When the Golgi is disassembled upon cellular stress, it signals to the nucleus to help relieve the stress[14]. Another effector from *S. flexneri*, IcsB, alters signaling by inducing lysine fatty acylation on an array of host proteins[15]. In contrast to the diverse host protein PTMs induced by pathogens, very little is known about whether host cells can reverse such modifications to fight back.

Lysine fatty acylation is an emerging PTM. Recently, multiple mammalian sirtuins and histone deacetylases (HDACs) were discovered to be able to catalyze the removal of lysine fatty acylation in vitro and in cells, which suggested that this PTM might be tightly regulated with important roles in cellular function[16–19]. However, only a few proteins were reported to possess lysine fatty acylation and the mammalian lysine fatty acyl transferase for most of the proteins remain unknown[18–23]. With the recent identification of IcsB as a robust fatty acyltransferase from *S. flexneri*[15], it is intriguing to hypothesize that host sirtuins and HDACs fight *Shigella* infection by counteracting the function of IcsB via their defatty-acylation activities.

Herein, we find SIRT2, a nicotinamide adenine dinucleotide (NAD$^+$)-dependent protein lysine deacylase, is transcriptionally induced by Golgi stress and serves as a potent defatty-acylase to counteract the action of IcsB during *Shigella* infection. These findings reveal an anti-bacterial role of SIRT2 and establish reversible lysine fatty acylation as an important PTM in the war between pathogenic bacteria and mammalian hosts.

## Results

### Golgi stress upregulates SIRT2 via CREB3
The project initially started when we were interested in finding out how SIRT2 is regulated as we believe knowing its regulation is crucial for understanding its biological function. Interestingly, we found that

[1]Department of Chemistry and Chemical Biology, Cornell University, Ithaca, NY 14853, USA. [2]Howard Hughes Medical Institute; Department of Chemistry and Chemical Biology, Cornell University, Ithaca, NY 14853, USA. [3]These authors contributed equally: Miao Wang, Yugang Zhang, and Garrison Komaniecki. ✉e-mail: hl379@cornell.edu

Brefeldin A (BFA) drastically increased SIRT2 protein levels in multiple cell lines (Fig. 1A–B). BFA is known to induce Golgi stress[24]. The ER stress inducers, Tunicamycin (Tm) and Thapsigargin (Tg), only slightly increased SIRT2 (Fig. 1C), which further confirmed the specificity of the stress response.

Quantitative real-time (RT)-PCR revealed that the *SIRT2* transcript level increased more than 50-fold under BFA treatment when compared to the vehicle control group (Fig. 1D). Moreover, 24 h of Actinomycin D (ActD) or cycloheximide (CHX) treatment abolished the protein upregulation. This was consistent with the observation that the

*SIRT2* transcript increased primarily during 12−24 h of BFA treatment and suggested SIRT2 mRNA stability or protein stability was not the key attributes (Fig. 1E–F). We also examined several other sirtuins and HDACs under Golgi stress. SIRT2 mRNA and protein level increase was the most drastic (Fig. 1G–H). Taken together, the data suggest SIRT2 is upregulated by Golgi stress through transcriptional control.

Next, we tried to identify the transcription factor responsible for *SIRT2* upregulation under Golgi stress. The Golgi apparatus is hypothesized to integrate and transduce stress stimuli to the nucleus[14]. One of the mechanisms is through the CREB3 transcription factor[25]. Thus,

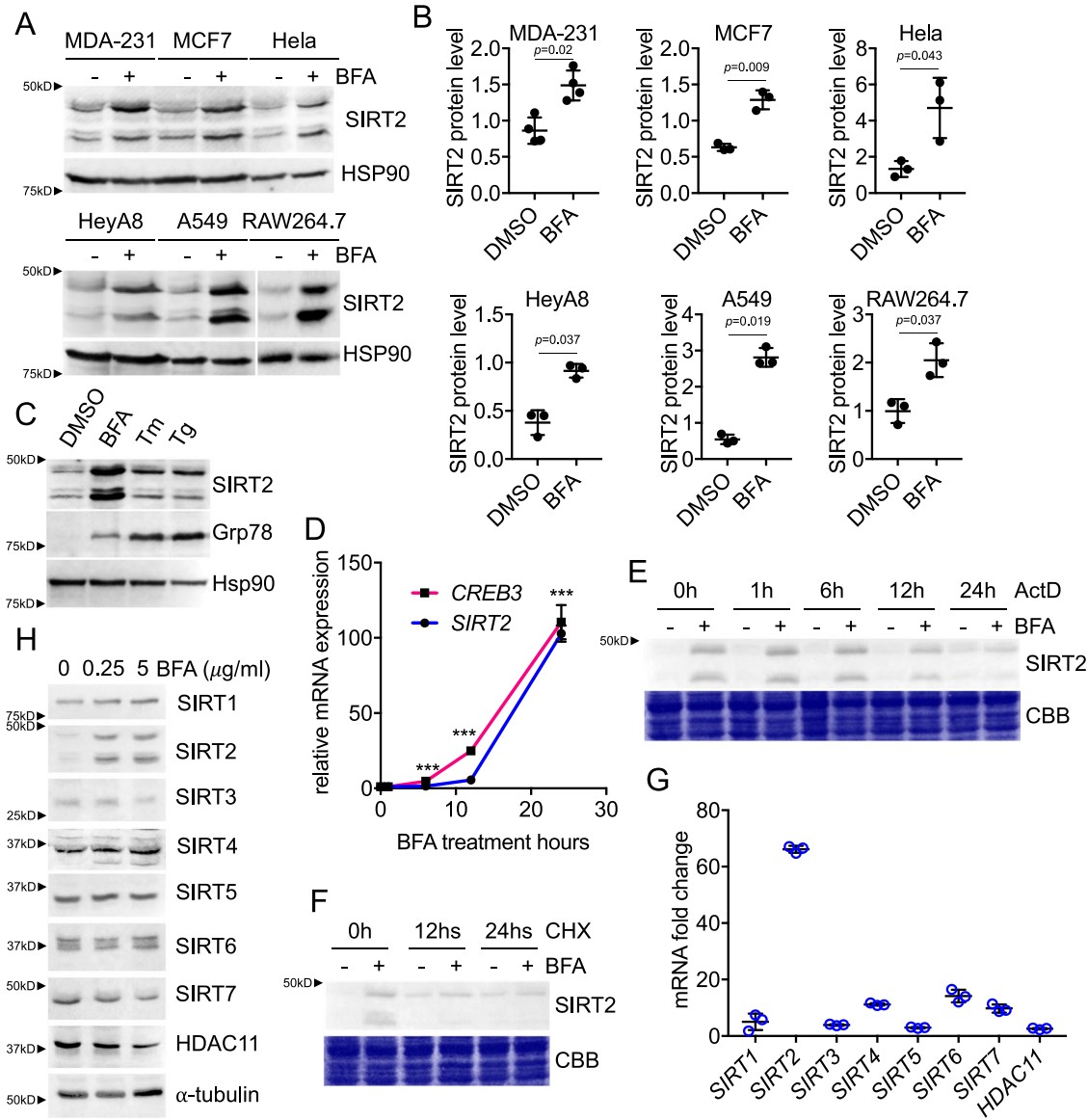

**Fig. 1 | Golgi stress transcriptionally upregulates SIRT2. A** Immunoblots for SIRT2 in different cell lines treated with 5 μg/ml BFA for 24 h. HSP90 blots are shown as loading controls. Representative images from three independent experiments are shown. **B** Quantification of relative SIRT2 protein levels in different cell lines, normalized to HSP90. Statistical evaluation was done by a paired two-tailed Student's *t*-test. *n* = 3 or 4 biological replicates. **C** In A549 cells, BFA treatment significantly increased SIRT2 protein level, while the ER stress inducers Tunicamycin (Tm) and Thapsigargin (Tg) only weakly induced SIRT2 expression. **D** RT-PCR analysis for *SIRT2* and *CREB3* mRNA levels in A549 cells treated with 5 μg/ml BFA for various incubation times. mRNA was normalized to 0 h. Statistical evaluation was done by an unpaired two-tailed Student's *t*-test. *n* = 3 biological replicates. **E** ActD chase analysis showing SIRT2 upregulation is due to mRNA upregulation, which happens primarily during 12–24 h. A549 cells were treated with or without

BFA at *t* = 0 h. During BFA treatment, ActD was added at *t* = 0, 12, 18, 23, 24 h. Cells were collected at *t* = 24 h and submit for western blot analysis. **F** CHX chase analysis showing SIRT2 upregulation is not due to protein stability. A549 cells were treated with or without BFA at *t* = 0 h. During BFA treatment, CHX was added at *t* = 0, 12, 24 h. At *t* = 24 h, cells were collected and submit for western blot analysis. **G** RT-PCR analysis for *SIRT1-7* and *HDAC11* mRNA level in A549 cells treated with 5 μg/ml BFA for 24 h. Fold change in mRNA was calculated by comparing samples with BFA treatment to control DMSO. *n* = 3 biological replicates. **H** Among all sirtuins and HDAC11 tested, SIRT2 protein level increase was the most drastic under Golgi stress. A549 cells were treated with BFA the indicated concentrations for 24 h before analyzed by western blot. α−tubulin blots were loading control. Data are represented as mean ± SEM. Statistical evaluation was done using unpaired two-tail Student's *t*-test. ****p < 0.001.*

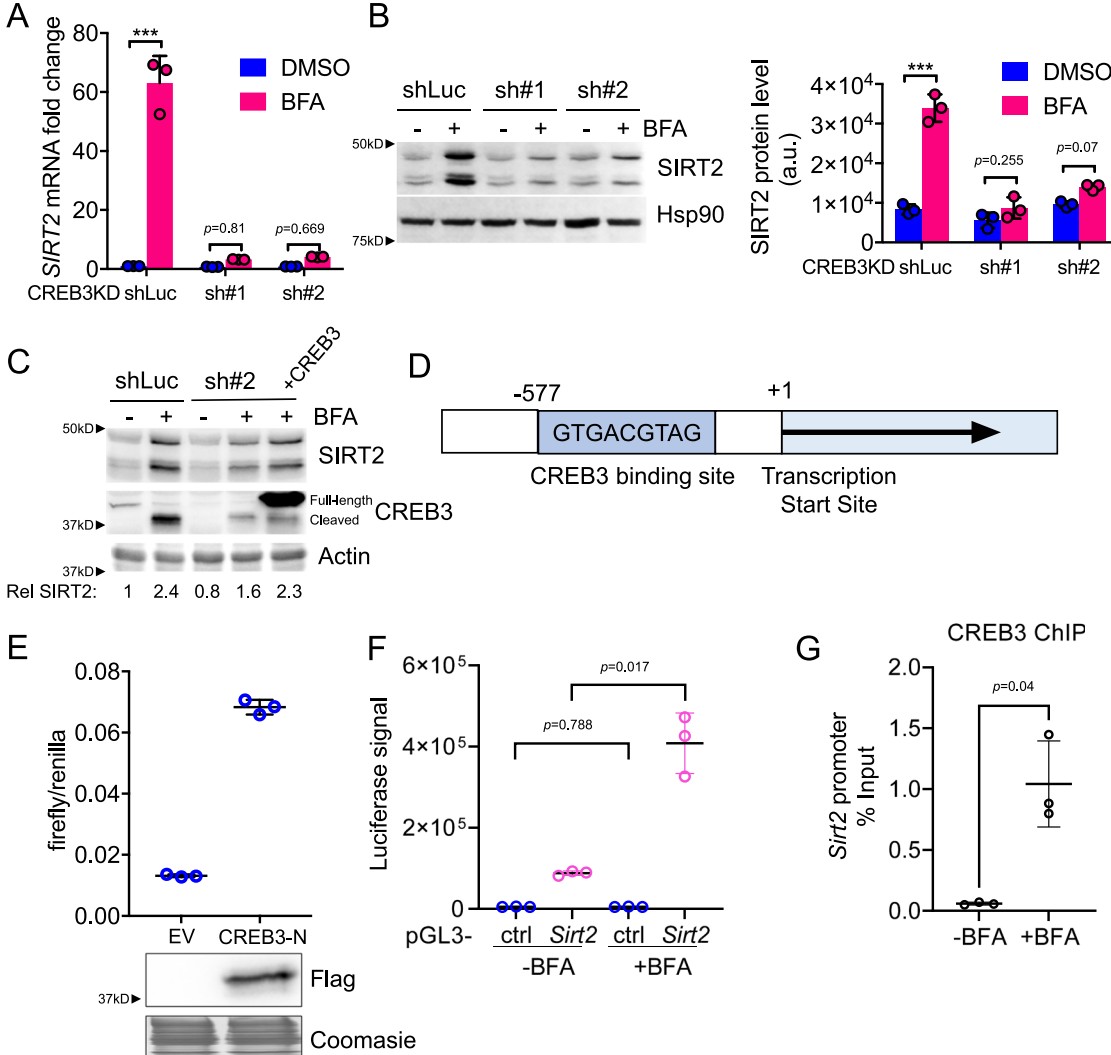

**Fig. 2 | CREB3 promotes SIRT2 transcription under Golgi stress. A** RT-PCR analysis for *Sirt2* mRNA in A549 CREB3 KD cells treated with or without BFA. mRNA is normalized to DMSO treated control (shLuc) cells. Statistical evaluation was done by two-way ANOVA. **B** Immunoblots for SIRT2 protein levels in A549 control (shLuc) and CREB3 KD cells treated with or without BFA. HSP90 blot was used as the loading control. The quantification of SIRT2 protein level, normalized to HSP90, is shown on the right. Statistical evaluation was done by two-way ANOVA. **C** Immunoblots for SIRT2 in control and CREB3 KD cells treated with BFA and rescued with CREB3 transfection. SIRT2 level relative to actin loading indicated below. **D** A schematic representation of SIRT2 promoter. The annotated regions are on Chromatin 19 complement [39,389,400-39,391,600], reference: GRCh37. **E** SIRT2-promoter driven firefly transcription in cells under Flag-tagged CREB3 N-terminal bZIP domain (CREB3-N) overexpression. The Renilla luciferase construct was used as an internal control. **F** Firefly luciferase transcription in cells treated with BFA. Cells were transfected with an empty pGL3-basic vector (ctrl) or pGL3-basic vector with 994 base pairs of SIRT2 5' UTR (*Sirt2*). **G** CREB3 chromatin IP followed by qPCR of the SIRT2 promoter using primers surrounding the putative CREB3 binding site. Results are from three separate experiments with three technical replicates each. Data are represented as mean ± SEM. Statistical evaluation was done using unpaired two-tail Student's *t*-test. ***$p < 0.001$.

we decided to test if CREB3 is the transcription factor controlling SIRT2 expression under Golgi stress.

We first confirmed that the *CREB3* transcript was increased after BFA treatment (Fig. 1D). To test if CREB3 upregulates SIRT2 under Golgi stress, *CREB3* stable knockdown cells were treated with BFA and other Golgi stress inducers. As expected, the upregulation of SIRT2 was diminished in *CREB3* knockdown cells at both protein and mRNA levels while overexpression of CREB3 rescued this effect (Fig. 2A–C, Supplementary Fig. 1A–B). Moreover, the *SIRT2* promoter contains a strong CREB3 binding site (Fig. 2D), and is responsive to the expression of the active (cleaved) form of CREB3 or BFA treatment (Fig. 2E–F, Supplementary Fig. 1C). CREB3 binding of the *SIRT2* promoter was further confirmed by CREB3 chromatin immunoprecipitation (ChIP)-polymerase chain reaction (PCR) which showed increased enrichment of the *SIRT2* promoter after BFA treatment (Fig. 2G). Thus, the increased SIRT2 expression under Golgi stress is through CREB3.

While trying to understand the physiological relevance of BFA-induced Golgi stress and the possible role of SIRT2 upregulation, it came to our attention that an intracellular pathogen, *Shigella flexneri*, is known to induce Golgi fragmentation to inhibit host cell secretion[13]. *Shigella* utilizes a type III secretion system (TTSS) to inject multiple effectors into host cells for successful invasion. One of the effectors, IpaJ, induces Golgi stress by proteolytic cleavage of N-terminal glycine myristoylated proteins, such as ADP-ribosylation factors (ARFs)[26]. To test if SIRT2 is upregulated by pathogen-induced Golgi stress, we overexpressed IpaJ in HEK293T and A549 cells. Indeed, SIRT2 protein level was upregulated (Fig. 3A and Supplementary Fig. 2E).

Infecting mouse bone-marrow-derived macrophage (BMDM) cells or mice with wildtype *S. flexneri* increased SIRT2 protein and mRNA levels while infection with the IpaJ deletion strain or IpaJ/VirA double deletion strain did not significantly change SIRT2 levels (Fig. 3B–C, Supplementary Fig. 2). Consistent with IpaJ upregulating SIRT2

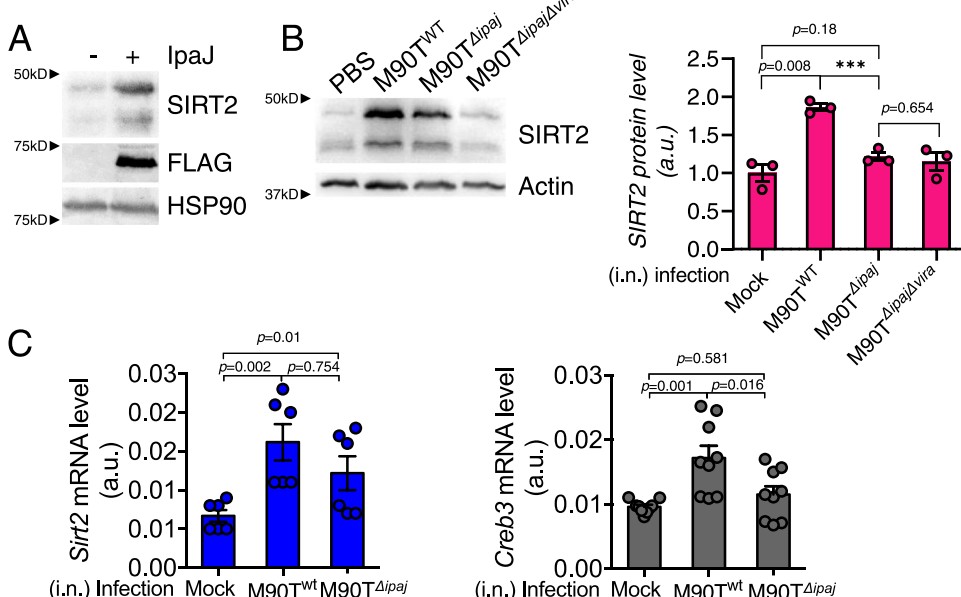

**Fig. 3 | Shigella infection upregulates SIRT2 through Golgi stress. A** SIRT2 protein level is induced by IpaJ, showing by immunoblots for SIRT2 protein levels in HEK293T cells with or without Flag-IpaJ overexpression. HSP90 is the loading control. **B** Immunoblots of SIRT2 in BMDMs infected with mock, wild-type or IpaJ deletion *S. flexneri* M90T. The BMDMs were derived from 6 to 8-week old wildtype C57BL/6 J mice. Quantification of SIRT2 protein levels (normalized to actin) in the immunoblots is shown on the right. Statistical evaluation was done by student *t*-test. **C** RT-PCR of *Sirt2* (left) and *Creb3* (right) in BAL cells from mice intranasally infected with mock, wildtype or IpaJ deletion *S. flexneri* M90T cells. mRNA level in each sample was normalized to internal control (*Actb*). Statistical evaluation was done by one-way ANOVA. Data are represented as mean ± SEM. *a.u., arbitrary unit*.

through CREB3, overexpression of IpaJ in CREB3 knockdown cells did not upregulate SIRT2 (Supplementary Fig. 2E). Infection of cells with enteropathogenic *E. coli*, as well as overexpression of EspG and VirA (Golgi stress-inducing bacterial toxins from enteropathogenic *E. coli* and *Shigella* respectively) also upregulated SIRT2 consistent with a conserved mechanism of SIRT2 regulation during bacteria-induced Golgi stress (Supplementary Fig. 2F-G). Overall, the data is consistent with the idea that *Shigella* infection upregulates SIRT2 via Golgi stress and suggest a potential role of SIRT2 in pathogen-host interaction.

## SIRT2 counteracts the action of IcsB

To identify the role of SIRT2 during *Shigella* infection, we focused on the enzymatic activity of SIRT2. SIRT2 can hydrolyze long-chain fatty acyl groups on protein lysine residues. Interestingly, another effector from *Shigella*, IcsB, is able to fatty acylate host proteins[15]. Therefore, we hypothesized that, upon *Shigella* infection, SIRT2 is upregulated to remove the PTM installed by IcsB as a host defense mechanism.

We first tested whether SIRT2 could reverse the fatty acylation added by IcsB in vitro. An alkyne-tagged long-chain fatty acid analog, Alk14, was used to metabolically label substrate proteins[27]. Several known IcsB substrate proteins were tagged with a flag tag, co-overexpressed with IcsB and immunoprecipitated from HEK293T cell that were treated with Alk14. Next, we incubated the isolated proteins with purified recombinant SIRT2 with or without NAD[+], and then conjugated the substrate proteins to BODIPY-azide using click chemistry to allow visualization of fatty acylation level by in-gel fluorescence. Hydroxylamine ($NH_2OH$) was used to remove cysteine palmitoylation. Remarkably, incubation of multiple IcsB substrate proteins, including Ras family (RalA, RalB, Rheb, RRas), Rho family (CDC42, Rac1), and an ESCRT component (CHMP5), with SIRT2 resulted in the removal of most of the lysine fatty acylation from the proteins in the presence of NAD[+] (Fig. 4A). This suggests that the substrate scope of IcsB and SIRT2 overlap significantly.

We next tested if SIRT2 overexpression could counteract the effect of IcsB in cells. In this experiment, SIRT2-H187Y (HY), a previously reported dead deacetylase but weak defatty-acylase, was used

as a control[18]. We co-overexpressed Flag-tagged IcsB substrate proteins, IcsB, and SIRT2 in HEK293T and examined the lysine fatty acylation on the substrate proteins by Alk14 labeling and in-gel fluorescence. As expected, co-expression of SIRT2 with multiple IcsB substrate proteins significantly decreased their lysine fatty acylation levels, whereas co-expression of SIRT2-HY had much less effect (Fig. 4B–C, Supplementary Fig. 3B, 4A–B). Similar trends were observed when HEK293T cells were infected with *S. flexneri* instead of ectopically overexpressing IcsB (Supplementary Fig. 3A). Additionally, IcsB substrates showed higher lysine fatty acylation levels in SIRT2 knockdown cells upon IcsB overexpression or *Shigella* infection (Supplementary Fig. 3C–D). The effect of SIRT2 knockdown was less pronounced compared to SIRT2 overexpression. We reasoned that the overexpression of SIRT2 more closely mimics *Shigella* infection where SIRT2 is upregulated by Golgi stress. Global protein fatty acylation labeling also showed that *Shigella* infection increased the labeling of several bands and SIRT2 knockdown further increased the labeling (Supplementary Fig. 4C).

It is reported that the fatty acylation on Rho GTPase inhibits the protein-protein interaction between Rho GTPases and RhoGDI[28], which regulates the membrane cycling and signaling of Rho GTPases[29,30]. We wanted to confirm these results and find out whether SIRT2 could rescue this effect. As shown in Fig. 5, lysine fatty acylation inhibits Rho GTPase and RhoGDI interaction, and this can be rescued by SIRT2 overexpression. Thus, SIRT2 works as a defatty-acylase to maintain Rho GTPase signaling during *Shigella* infection.

We tested other enzymes that can remove long-chain fatty acyl groups from protein lysine residues and found that HDAC11, but not SIRT6 or SIRT7, could also decrease IcsB substrate fatty acylation (Supplementary Fig. 3E), suggesting that HDAC11 could also be involved[16,17,19]. However, HDAC11 is not regulated by Golgi stress (Fig. 1G), which is different from SIRT2. The strong activity of SIRT2 in comparison to SIRT6 and SIRT7 could be explained by the fact that SIRT2 has more efficient activity in vitro and it lacks substrate peptide sequence selectivity[17,31,32]. Also, SIRT2 is the only sirtuin with a predominate cytosolic localization, which allows the interaction with IcsB

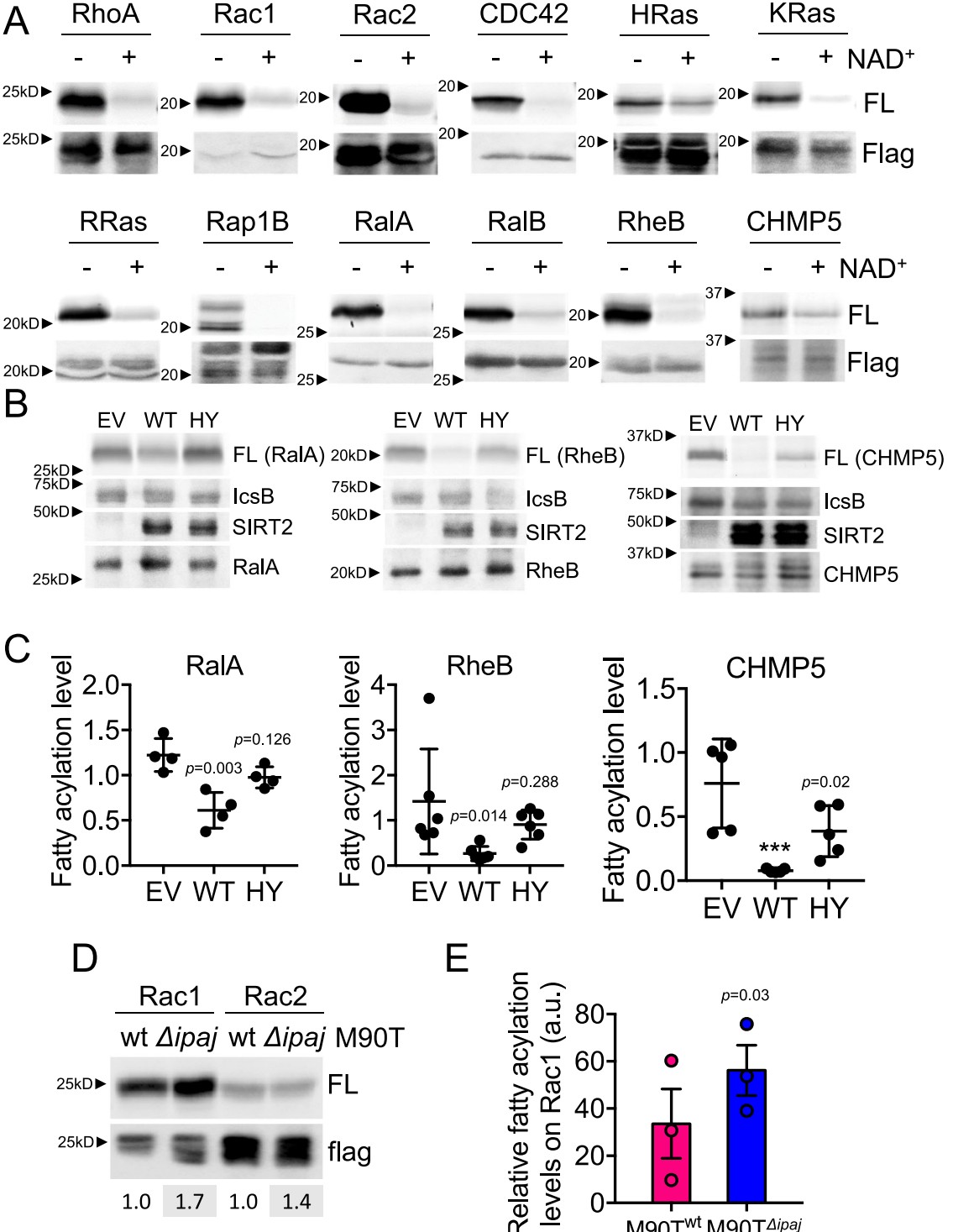

**Fig. 4 | SIRT2 is a potent lysine defatty-acylase that counteracts IcsB. A** In-gel fluorescence detection of lysine fatty acylation of Flag-tagged IcsB substrate proteins treated with 5 μM of SIRT2, with or without 1 mM of NAD⁺ in vitro. **B** In-gel fluorescence detection of lysine fatty acylation of Flag-tagged IcsB substrates in HEK293T cells that were also transfected with Flag-tagged IcsB and SIRT2. Representative images from at least three independent experiments are shown. EV, empty vector. WT, SIRT2 WT. HY, SIRT2 HY. FL, fluorescence (indicative of lysine fatty acylation level on substrate proteins). Flag, anti-Flag immunoblot (indicative of input level of substrate proteins). **C** Quantification of the lysine fatty acylation levels in **B**. $n = 4–6$ biological replicates. **D** In-gel fluorescence detection of lysine fatty acylation of Flag-tagged Rac1 and Rac2 in HEK293T cells that were infected with equal number of WT or IpaJ deletion *S. flexneri* M90T cells. FL, fluorescence (lysine fatty acylation level). Flag, anti-Flag immunoblots. **E** Quantification of the lysine fatty acylation levels detected in **D**. $n = 3$ biological replicates. Data are represented as mean ± SEM. Statistical evaluation was done using one-way ANOVA. ***$p < 0.001$.

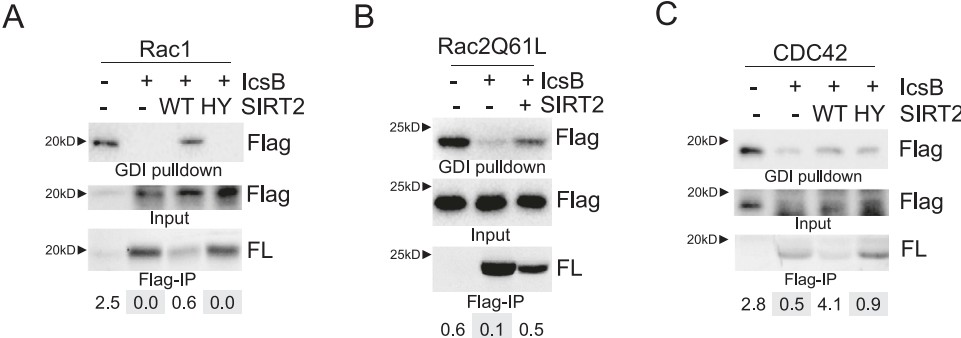

**Fig. 5 | SIRT2 -catalyzed defatty-acylation promotes Rho GTPases and RhoGDI interaction.** The disruption of Rho GTPase binding to RhoGDI by IcsB is rescued by SIRT2. HEK293T cells were transfected with plasmids encoding indicated proteins: IcsB, SIRT2 WT or HY mutant, and Flag-tagged Rac1 (**A**), Rac2 (**B**) or CDC42 (**C**). Cell lysates for each sample were subjected to GST-RhoGDI pulldown assay (shown in the first panel) and lysine fatty acylation labeling assay (shown in the last panel). The quantification of GDI enrichment of Rho GTPases, obtained by dividing Flag-Rho GTPase signal in GDI pulldown with Flag-Rho GTPase in the input (second panel), is shown at the bottom.

substrates. With the same multiplicity of infection (MOI), lysine fatty acylation levels of IcsB substrate proteins are moderately higher in cells infected with IpaJ deletion *S. flexneri* (Fig. 4D–E), consistent with the idea that SIRT2 is upregulated by IpaJ-induced Golgi stress. Taken together, this highlights the uniqueness of SIRT2 as the defatty-acylase and Golgi stress-mediated SIRT2 regulation.

### SIRT2 limits *Shigella* autophagosome escape

Notably, the in-cell labeling results suggest that SIRT2 prefers certain substrates (Supplementary Fig. 3B). CHMP5, an ESCRTIII component, was one of the preferred substrates as it showed more than 9-fold decrease in fatty acylation level upon SIRT2 overexpression. CHMP5 K7R, a mutant that lacks the primary IcsB modification site, only had some very weak fatty acylation that could be removed by SIRT2, suggesting SIRT2 is able to remove fatty acylation from redundant sites (Supplementary Fig. 3F).

Host cells use autophagy to target intracellular bacteria and to restrict bacterial growth[33]. IcsB is important for *Shigella* to escape host autophagy by modifying CHMP5[15,34]. We hypothesized that SIRT2, working against IcsB, would promote the innate immunity and suppress *Shigella* escaping from autophagosomes. To test this hypothesis, SIRT2 knockout (*Sirt2*[-/-]) and wildtype (*Sirt2*[+/+]) mouse embryonic fibroblast (MEF) cells were infected with *S. flexneri*. Bacteria that were trapped in autophagosomes were visualized by staining LC3, an autophagosome-specific marker. Consistent with previous reports, more *Icsb* deletion (*Δicsb*) *Shigella* was trapped in autophagosomes than wildtype *Shigella* (Fig. 6). Excitingly, the percentage of *Sirt2*[-/-] MEF cells containing LC3 decorated wildtype *Shigella* was significantly less than those of *Sirt2*[+/+] MEF cells (Fig. 6). This is further confirmed by LC3 turnover assay (Supplementary Fig. 5). The data supports that SIRT2 limits *Shigella* autophagosome escape.

### SIRT2 restricts *Shigella* infection in cells and in mice

To further test if SIRT2 is able to restrict proliferation of *Shigella*, we performed a gentamycin killing assay to examine the intracellular *Shigella* growth. Equal number of *Sirt2*[+/+] and *Sirt2*[-/-] MEF cells were infected with equal number of *S. flexneri* for 10 min before treated with gentamycin to kill extracellular bacteria. The intracellular bacteria number was measured by recoverable colony formation units (CFU) to give the MOI. This assay revealed that *Shigella* survived better without SIRT2, at both early and later time points (Fig. 7A). Similar trends were also observed in *Sirt2*[-/-] MEF cells that re-express empty vector or SIRT2, as well as in Caco2 and A549 control or SIRT2 knockdown cells (Fig. 7B–C, Supplementary Fig. 6A–B). CREB3 KD cells have increased bacterial load after infection which can be partially rescued by WT but not HY SIRT2, suggesting that regulation of SIRT2 is a key function for

CREB3 during *Shigella* infection (Supplementary Fig. 6C-D). The protective effect of SIRT2 is impaired against *S. flexneri* IpaJ deletion strain, suggesting the relevance of Golgi stress mediated SIRT2 regulation (Fig. 7D). Furthermore, *Sirt2*[+/+] MEF cells survived better under *Shigella* infection or BFA treatment conditions than *Sirt2*[-/-] cells (Supplementary Fig. 6E–F).

We also examined the effect of *Sirt2* in wildtype and *Sirt2* knockout C57/B6J mice. Six to eight-week old mice were intranasally administrated with *S. flexneri*, closely monitored, and sacrificed after 1–3 days. *S. flexneri* present in the lung were quantified. Consistently, there were more *S. flexneri* in *Sirt2*[-/-] mice, which further confirmed our model that SIRT2 protects against *Shigella* infection (Fig. 7E, Supplementary Fig. 7). We observed drastically reduced physical activity in *Sirt2*[-/-] mice comparing with *Sirt2*[+/+] mice after bacterial infection (attached video in supplementary information). Moreover, there were lower circulating cytokine (TNF-α and IL-17) levels in *Sirt2*[-/-] mice (Fig. 7F), suggesting that SIRT2 promotes the immune response against *S. flexneri* infection. Overall, the data demonstrate SIRT2 restricts *Shigella* infection both in cells and in vivo.

To validate if the effect of SIRT2 was through counteracting the effect of IcsB, we infected the same number of *Sirt2*[+/+] and *Sirt2*[-/-] MEF cells with same number of wildtype and *Δicsb S. flexneri*. Consistently, wildtype *S. flexneri* proliferated much better in *Sirt2*[-/-] cells than in *Sirt2*[+/+] cells (Fig. 8A). In contrast, *Δicsb S. flexneri* proliferated at similarly low levels in both *Sirt2*[+/+] and *Sirt2*[-/-] cells, suggesting SIRT2 counteracts IcsB to limit *Shigella* infection (Fig. 8A). Interestingly, wildtype and *Δicsb S. flexneri* proliferated similarly in *Sirt2*[+/+] MEF cells, suggesting that SIRT2 is able to fully counteract IcsB (Fig. 8A). The same trend was also observed in vivo (Fig. 8B). In intranasally infected mice, we observed similar MOI for *Δicsb S. flexneri* in *Sirt2*[+/+] and *Sirt2*[-/-] mouse lungs, but significantly higher MOI of wildtype *S. flexneri* in *Sirt2*[-/-] than in *Sirt2*[+/+] mouse lungs (Fig. 8B). To further confirm the SIRT2/IcsB axis, we generated *Δicsb S. flexneri* rescued by either WT or catalytic dead C306A IcsB. When infected with WT IcsB *S. flexneri*, *Sirt2*[-/-] mice had much more bacterial colonization. *Sirt2*[+/+] and *Sirt2*[-/-] mice showed no statistically significant difference in colonization by C306A *S. flexneri* (Fig. 8C). Similar, though less pronounced trends were observed in intraperitoneally infected mice (Supplementary Fig. 8), which is possibly due to complication from severe inflammation reactions[35]. These results demonstrate the in vivo anti-bacterial role of SIRT2 as a defatty-acylase to counteract the action of IcsB.

## Discussion

Emerging studies suggest that pathogens employ PTMs to modulate host protein functions to facilitate pathogenesis[36]. Such PTMs are often highlighted with novel enzymatic reactions and high reaction

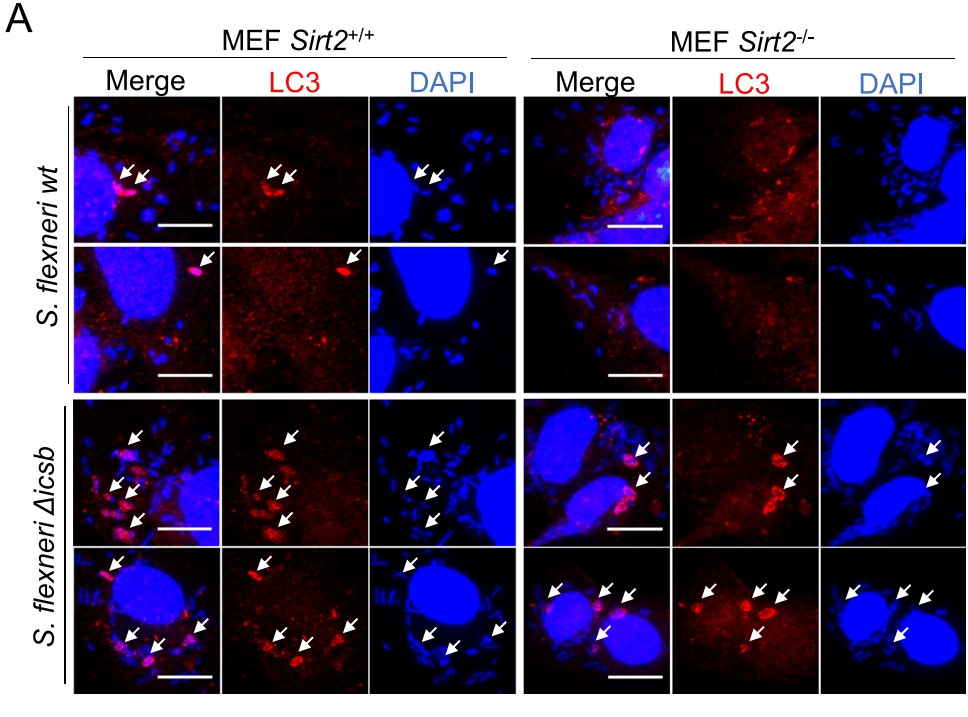

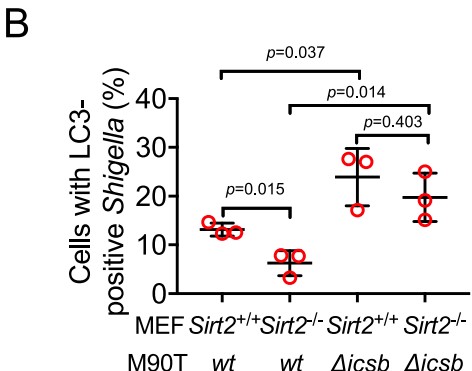

**Fig. 6 | SIRT2 suppresses Shigella autophagosome escape. A** Effect of SIRT2 knockout on *S. flexneri* autophagosome escape. Representative images of MEF cells infected with *S. flexneri* from three biological replicates are shown. Scale bar: 15 μm. White arrow: LC3-positive *Shigella*. **B** Quantitation of data shown in **A**. Percentage of MEF cells containing LC3-positive *shigella* was quantified. *n* = 3 biological replicates. Data is shown as mean ± SEM with >100 infected MEF cells counted for each experiment. Statistical evaluation was done using two-way ANOVA. Data are represented as mean ± SEM.

efficiency. One of the PTMs, protein fatty acylation, regulates the association of proteins with membranes, and is important for membrane trafficking, protein-protein interaction, and signal transduction[37]. Fatty acylation on lysine residues, however, is less well studied. Multiple mammalian lysine defatty-acylases were identified[16,18,19,22,38], which strongly suggested the physiological relevance of this PTM. However, very little is known on mammalian lysine fatty acyltransferases[23], which has greatly limited the study of the function of this PTM. Our findings demonstrate that reversible lysine fatty acylation plays an important role in pathogen-host interaction, highlighting the importance of lysine fatty acylation in innate immunity.

SIRT2 was initially recognized as a lysine deacetylase, with its cellular function mostly attributed to its deacetylation activity[39]. Many in vitro studies suggested that SIRT2 can also catalyze lysine defatty-acylation[40]. However, the first physiological de-fatty acylation substrate, K-Ras4a, was only recently identified[18,41]. Our findings here thus provide key insights that will help understand the physiological significance of SIRT2's defatty-acylation activity and expands its substrate scope. While we believe the defatty-acylation activity of SIRT2 plays a predominate role in fighting *Shigella* infection, we do not rule out that the deacetylation activity could also be important.

As the hub of intracellular membrane trafficking, the Golgi apparatus possess a unique structure where the stacks are interconnected into a compact ribbon. The ribbon undergoes dynamical membrane fission, fusion, and cargo transport through cisternal maturation[42]. Upon certain cellular events such as mitosis and stresses conditions, the ribbon can undergo distinct disassembly processes, which will lead to disruption of Golgi integrity and stress stimuli transduction to the nucleus[14]. Our study revealed an important function of Golgi stress pathway in fighting *Shigella* infection. Through Golgi stress, cells sense *Shigella* infection and activate SIRT2 via the CREB3 transcription factor to enhance innate immunity. Among all of the known lysine defatty-acylases, SIRT2 is the most responsive to Golgi stress induced by pathogens. This highlights the unique role of SIRT2 in stress response.

Previously, SIRT2 was reported to be hijacked by other pathogens to work as deacetylase and to promote pathogen invasion[43–45].

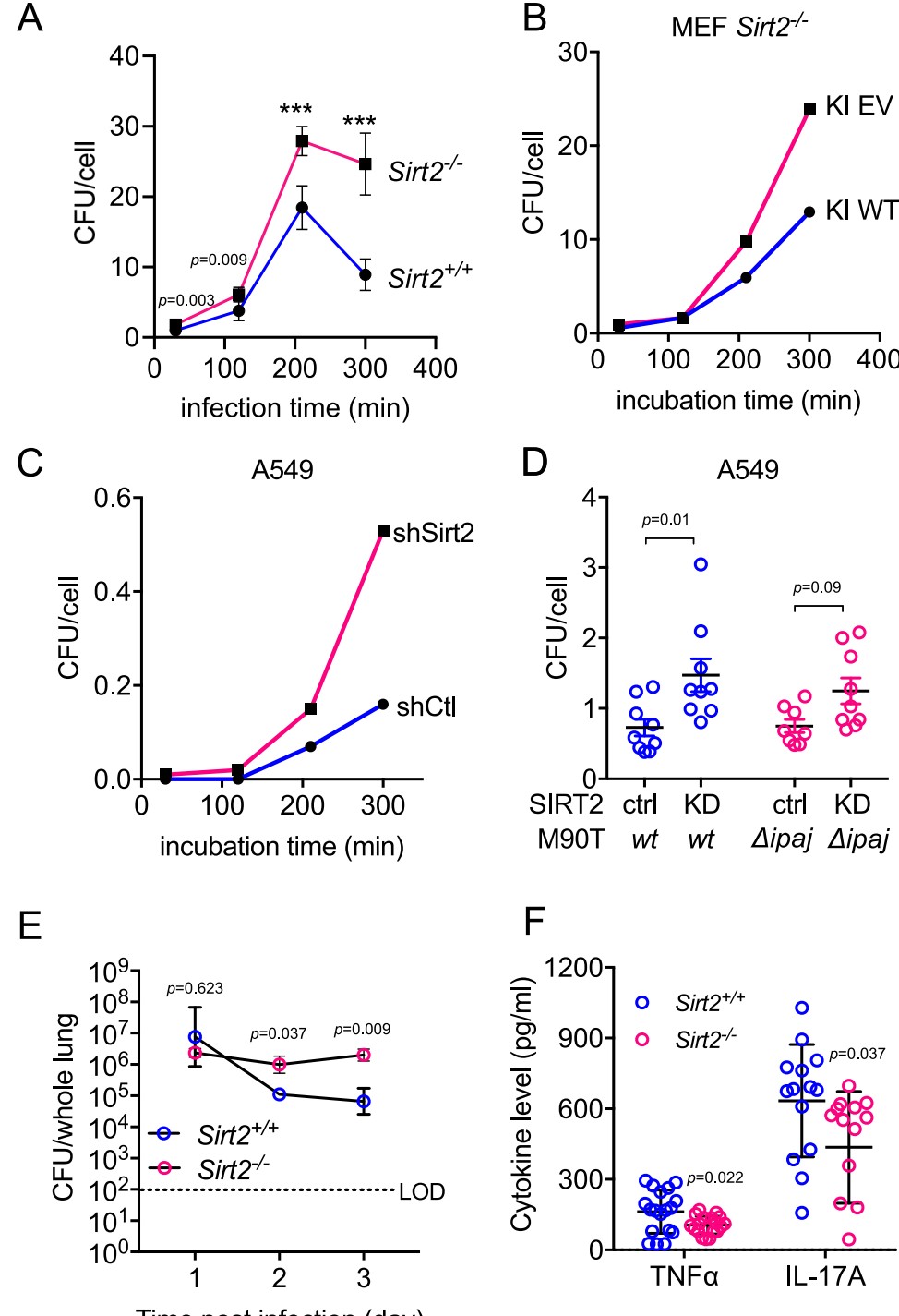

**Fig. 7 | SIRT2 restricts Shigella infection in cells and in mice model. A** Effect of *Sirt2* knockout on *S. flexneri* intracellular proliferation analyzed by the gentamycin killing assay. MEF *Sirt2*[+/+] and *Sirt2*[−/−] cells were infected with *S. flexneri* for 10 min. Intracellular *S. flexneri* number and MEF cell number were counted at indicated time points to get MOI value. Data are presented as mean ± SEM with three biological replicates (two technical replicates). Statistical evaluation: unpaired two-tailed Student's *t*-test. **B**, **C** Effect of SIRT2 knockout on *S. flexneri* intracellular proliferation. MOI were determined after 10 min of infection and the indicated time of incubation in the presence of 50 μg/ml gentamicin in *Sirt2*[−/−] MEF with empty vector or SIRT2 knock-in (KI, **B**) and in A549 with control or SIRT2 knockdown (**C**) Data are presented as mean ± SEM (the SEM is very small and hard to see in the figure) with three biological replicates (two technical replicates). Statistical evaluation: unpaired two-tailed Student's *t*-test. **D** A549 control and SIRT2 KD cells were infected with wildtype and IpaJ deletion *S. flexneri* M90T cells for 10 min, then treated with gentamycin to kill extracellular bacteria. Intracellular *S. flexneri* number and mammalian cell number were counted to get MOI. Data are presented as mean ± SEM with nine biological replicates. Statistical evaluation: two-way ANOVA. **E** Recoverable CFU in lung homogenates from 6 to 8-week old *Sirt2*[+/+] and *Sirt2*[−/−] C57BL/6 J mice intranasally infected with 1 million *S. flexneri* M90T wildtype strain. Statistical evaluation: unpaired two-tail Student's *t*-test. Data are presented as mean ± SEM with three mice per group. LOD, limit of detection. **F** Cytokine profile in *Sirt2*[+/+] and *Sirt2*[−/−] mice lung homogenates after intranasal infection with 1 million *S flexneri* for 2 days. *n* = 18 (TNFα) or 14 (IL−17A) biological replicates. Cytokine levels in PBS-challenged mice fell below the detection limit for the experimental setup so were not included in the graph. Data are represented as mean ± SEM. Statistical evaluation: unpaired two-tail Student's *t*-test. ****p* < 0.001.

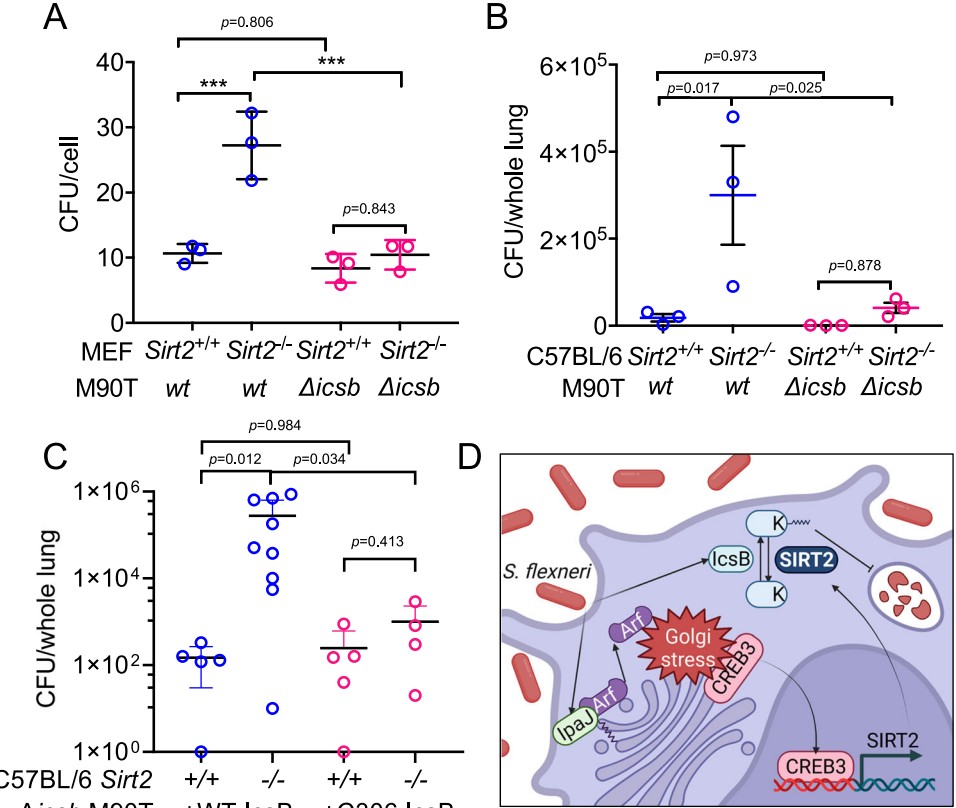

**Fig. 8 | SIRT2 restricts Shigella infection in cell and in vivo by counteracting IcsB. A** Effect of SIRT2 knockout and IcsB deletion on *S. flexneri* intracellular proliferation. MOI were determined after 10 min of infection and 6 h of incubation in the presence of 50 µg/ml gentamicin. Data are represented as mean ± SEM with three biological replicates. Statistical evaluation was done by two-way ANOVA. **B** CFU in lung homogenates from *Sirt2⁺ᐟ⁺* and *Sirt2⁻ᐟ⁻* C57BL/6 J mice infected with wildtype or IcsB deletion *S. flexneri* M90T for 3 days. Data are presented as mean ± SEM with three mice per group.

Statistical evaluation was done using two-way ANOVA. **C** CFU in lung homogenates from *Sirt2⁺ᐟ⁺* and *Sirt2⁻ᐟ⁻* C57BL/6 J mice infected with wildtype or IcsB deletion *S. flexneri* M90T rescued with WT or C306A IcsB for 3 days. *n* = 4–9 biological replicates. Data are presented as mean ± SEM with at least three mice per group. Statistical evaluation was done using two-way ANOVA. **D** Model depicting SIRT2 as a Golgi stress response protein that limits *Shigella* pathogenesis by counteracting Shigella-mediated host protein lysine fatty acylation. ***p < 0.001.

Here our study discovered an anti-bacterial role of SIRT2 by counteracting the fatty acyltransferase IcsB from *S. flexneri*. Many other pathogen fatty acyltransferases have been identified or predicted. For example, MARTX from *Vibrio cholerae* shares sequence similarities with cholera toxin, also disrupts Golgi apparatus in a pattern that is reminiscent of the actions of BFA[47]. This suggests that the mechanism of SIRT2's anti-bacterial role is potentially broader. One limitation of our study is that we used a mouse lung infection model for *Shigella*. While this model is useful for deciphering local immune protection, it does not mimic bacillary dysentery, the usual disease caused by *Shigella* infection in humans.

In addition to lysine fatty acylation, pathogenic bacteria use many other PTMs to help infect host cells. Here we reveal lysine fatty acylation as a pathogen-induced PTM that can be efficiently removed by host enzymes. This suggests that other pathogen-induced PTMs may also be reversed by host enzymes and studying them may provide insights into mechanisms of innate immune responses.

## Methods
### Reagents, antibodies and plasmids
Tunicamycin, Thapsigargin, cycloheximide were purchased from Sigma-Aldrich (St. Louis, MO); Actinomycin D and gentamicin were from Thermo Fisher. Brefeldin A was from Cell Signaling Technology (Danvers, MA). NAD⁺ was purchased from VWR (Radnor, PA). Alk14 and BODIPY-azide were synthesized as previously described[27].

The following antibodies were purchased from Cell Signaling Technology (Danvers, MA): SIRT2 (D4050, #12650, used at 1:1000), PARP (#9542, used at 1:1000), LC3 (#3868, used at 1:1000), and anti-rabbit IgG HRP-linked Antibody (#7074, used at 1:2000). The following antibodies were from Santa Cruz: β-actin (C-4, sc-47778, used at 1:2000), GRP78 (H-129, sc-13968, used at 1:500). Anti-Flag M2 antibody conjugated with horseradish peroxidase (A8592, used at 1:3000) and anti-flag M2 affinity gel (A220) were purchased from Sigma. Anti-CREB3 (ab180119, used at 1:2000) was from Abcam (Cambridge, MA). Cy3-conjugated goat anti-rabbit IgG (H + L) was purchased from Thermo Fisher Scientific (A10520, used at 1:1000).

*IcsB* gene was synthesized by Integrated DNA Technologies (Coralville, IA) after codon optimization from *Shigella flexneri* 5a plasmid pWR100 193718 – 195202 with N-terminal Triple Flag-tag. The gene fragment was later cloned into pCMV4a between EcoRI and XhoI using primers 5'- AGT CAG GAA TTC ACC ATG GAT TAC AAA GAT CAC G –3' and 5'- AGT CAG CTC GAG CTA AAT ATT TGA ATG GGA GTT GTT GA –3' (the restriction sites are underlined).

*IpaJ* was synthesis by Integrated DNA Technologies and was cloned into pEGFP-C1-FLAG-GFP-Ubl4A-C (Addgene #86923) between BglII and SalI with N-terminal Flag-tag followed by EGFP using primers 5'- AGT CAG AGA TCT ATG TCG GAA CAA CGG AAG –3' and 5'- AGT CAG GTC GAC TTA CAA AGC CTC ATT AGT TAT AAC TAT GGA –3'. pEGFP-C1-FLAG-GFP-Ubl4A-C(90-157) was a gift from Yihong Ye (Addgene plasmid # 86923; http://n2t.net/addgene: 86923; RRID:Addgene_86923)[48]. For lentivirus generation, *IpaJ* was cloned into pCDH-SIRT2-Flag (Addgene # 102624) between EcoRI

and XhoI with N-terminal Flag-tag using primers 5'- AGT CAG GAA TTC ATG TCG GAA CAA CGG AAG −3' and 5'- AGT CAG CTC GAG TTA CAA AGC CTC ATT AGT TAT AAC TAT GGA −3'. pCDH-SIRT2-Flag was previously cloned[49].

Human *CHMP5* was amplified from HEK293T cDNA library and cloned into pCMV4a between EcoRI and XhoI with C-terminal Flag-tag using primers 5'- AGT CAG GAA TTC ACC ATG AAC CGA CTC TTC GGG −3' and 5'- AGT CAG CTC GAG TGA AGC AGG GAT CTG TGG −3'. CHMP5 K7R was generated through site directed mutagenesis using primers 5'- AAC CGA CTC TTC GGG AGA GCG AAA CCC AAG GCT −3' and 5'- TCT CCC GAA GAG TCG GTT CAT GGT GAA TTC CTG −3' (the mutation sites are underlined).

Human RhoGDI was amplified from HEK293T cDNA library and cloned into pGEX4T3 between BamHI and EcoRI using primers 5'- AGT CAG GGA TCC ATG GCT GAG CAG GAG C −3' and 5'- AGT CAG GAA TTC TCA GTC CTT CCA GTC CTT C −3'. RhoA, Rac1, Rac2, CDC42, Has, Kras, RRas, Rap1b, RalA, RalB, RheB were generated using a standard PCR cloning strategy.

## Cell culture, transfection and transduction

Human MCF7, Hela, MDA-MB-231, A2780, and HeyA8 were grown in DMEM media (Invitrogen) supplemented with 10% (v/v) heat-inactivated fetal bovine serum (FBS; Invitrogen, Carlsbad, CA). MEF cells were cultured in DMEM supplemented with non-essential amino acids and 15% (v/v) heat-inactivated FBS. A549 cells were cultured in RPMI-1640 (Invitrogen) with 10% (v/v) heat-inactivated FBS. HEK293T were cultured in DMEM supplemented with 10% (v/v) heat-inactivated calf serum (Sigma).

For transient overexpression in HEK293T cells, mammalian expression vectors were transfected using PEI MAX 40 K (Poly-Sciences) according to the manufacturer's protocol. Empty vector was transfected as negative control. To transduce A549 for overexpression or knockdown, lentiviral infection was performed as previously described[18]. The pLKO.1-puro lentiviral shRNAs constructs for Luciferase and human CREB3 were purchased from Sigma-Aldrich. Luciferase shRNA (SHC007), CREB3 shRNA1 (TRCN0000020342), CREB3 shRNA2 (TRCN0000020343) were used.

*Sirt2* WT and KO MEF, *Sirt2* KO MEF with stable *SIRT2* re-overexpression and *SIRT2* stable knockdown A549 and HEK293T cells were generated as previously described[18].

For ActD and CHX chase experiments, A549 cells were treated with or without BFA at *t* = 0 h. During BFA treatment, ActD was added at *t* = 0, 12, 18, 23, 24 h; or CHX was added at *t* = 0, 12, 24 h. at *t* = 24 h, cells were collected and submit for western blot analysis.

For IpaJ transfection of control and ShCREB3 A549 cells, cells were kept at less than or equal to 50% confluence for at least three passages to ensure maintenance of CREB3 knockdown and equal level of SIRT2 (SIRT2 levels may be regulated by confluence). In 6-well plates, control cells were transfected with 2 µg IpaJ-flag and ShCREB3 cells transfected with 0.7 µg IpaJ-flag to ensure equal expression levels of IpaJ. Cells were collected for immunoblot analysis 48 h after transfection.

## Bacterial strains and infection

*Shigella flexneri* 5a M90T is a kind gift from Prof. Neal M. Alto from UT Southwestern. ΔIcsB M90T was generated using the lambda red recombineering system[13]. Briefly, M90T containing pKD46 was transformed with KanR cassette from pKD4. Kanamycin resistant colonies were selected and cured for pKD46 to generate IcsB <>kan. The resistance was later removed with pCP20, resulting in a traceless gene deletion. The KanR cassette was amplified using primers 5'- ACA TCC CCA CAA TCA CCA AGT AAT GGA GAG TTA ATA AAG TGT GTA GGC TGG AGC TGC TTC −3' and 5'- AAA GTT TAT CAT ATA GTT TGC GAC ACA TTT CTA TGG CCT TAT GGG AAT TAG CCA TGG TCC −3' (the external overlap with IcsB was underlined).

## Gentamycin killing assay

*S. flexneri* infection was done as previously described[13]. Briefly, ~1 × 10⁶ mammalian cells were seeded into 6-well dishes. *S. flexneri* M90T was inoculated overnight at 30 °C in Luria-Bertani (LB) broth (Fisher, Hampton NH) with shaking. Before infection, M90T culture was back diluted 1:100 in fresh LB broth and incubated at 37 °C with shaking until the OD600 reached 0.4. The bacteria were further incubated with 0.03% Congo Red in PBS for 15 min before adding 20 µL bacterial suspension to mammalian cells for an MOI of ~8. The infection was facilitated by centrifugation at 1000 g at room temperature for 10 min. After 10 min, the mammalian cells were washed with PBS five times and incubated in fresh media containing 50 µg/ml gentamycin. At the indicated time points (30 min to 5 h), mammalian cells were washed with PBS three times, collected and counted under a light microscope. The cells were then lysed with 0.5% Triton X-100 in PBS and the cell lysates were plated on LB agar and colonies forming units (CFU) were counted after overnight incubation at 37 °C.

## Infection of mammalian cells with enteropathogenic *E. coli*

Enteropathogenic *E. coli* (EPEC) strain JPN15 (serotype O127:H6) was purchased from Bei Resources (NR-50517). EPEC were cultured overnight in LB media at 37 °C. The following day, the EPEC culture was diluted 1:200 in LB media and grown to OD$_{600}$ = 0.4. EPEC was added to a 70% confluence well of A549 cells grown in a 6-well plate at an MOI of 0.5 for 18 h before collection.

## Western blot analysis

The proteins of interest were detected using enzyme-linked fluorescence (ECL Plus; Pierce Biotechnology Inc.) and visualized using the Typhoon 9400 Variable Mode Imager (GE Healthcare, Piscataway, NJ). Quantification of the western blots was done using ImageJ software.

## RT-PCR analysis of mRNA levels

Total mRNA was extracted using the RNeasy Mini Kit (Qiagen, CA, USA) according to the manufacturer's instructions and then reverse transcribed to cDNA using SuperScript Vilo cDNA Synthesis Kit (Thermo Fisher). Real-time PCR were performed on QuantStudio™ 7 Flex Real-Time PCR System using SYBR™ Green PCR Master Mix (Applied Biosystems) and the primers shown in Table S1.

## Luciferase assay for Sirt2 promoter activity

The 5' Sirt2 promoter corresponding to the region −1 to −994 nucleotides upstream of the Sirt2 transcription start site was cloned into the pGL3-basic vector (Promega) to create pGL3-994. Firefly/renilla luciferase ratio was determined by co-transfecting A549 cells with pGL3-994 and pGL4.73 (Renilla luciferase) for 24 h. Cells were either co-transfected with N-terminal flag tagged CREB3 (1-220) or treated with 5 µg/mL BFA for an additional 24 h. SIRT2 promoter was cloned into pGL3-basic using Gibson cloning using the primers 5'- GTG CTA GCC CGG GCT CGA GAT CTG CGA TCT AAG TAC CAC AGT TCT AAC AGA AGT CTC AGG −3' and 5'- AGA ATG GCG CCG GGC CTT TCT TTA TGT TTT GGC CGT CTT CCA TGG GCG CGG TG −3'. CREB3 (1-220) was cloned into pCMV5 between EcoRI and XhoI using primers 5'- AGT CAG GAA TTC ATG GAC TAC AAA GAC GAT GAC GAC AAG ATG GAG CTG GAA TTG GAT G −3' and 5'- AGT CAG CTC GAG CTA TGA TAT CTC AAT CAC CAT GGC −3'. Dual-luciferase activity was then measured using the Promega Dual-Luciferase® Reporter Assay (E1910) according to manufacture specifications. To determine BFA-induced luciferase signal for SIRT2 promoter, A549 cells were transfected with pGL3-basic or pGL3-994 for 24 h then treated with 5 µg/mL BFA for an additional 24 h. Cells were suspended in PBS to determine cell density and then processed according to manufacturer recommendations (Promega E1910). Firefly luciferase signal was normalized to cell number.

## Chromatin IP

Approximately $1.5 \times 10^7$ A549 cells in a 15 cm dish were treated with 5 µg/mL BFA or DMSO for 24 h. Cells were crosslinked with 1% formaldehyde for 10 min and processed according to ChIP-IT® Express Enzymatic Magnetic Chromatin Immunoprecipitation Kit (Active Motif, 53009) manufacturer recommendations using an α-CREB3 antibody (Proteintech, 11275-1-AP). Chromatin inputs and eluted ChIP samples were analyzed via qPCR using the primers 5'- AGA CTC TAG ACC CCT GGT GG −3' and 5'- TTG GAG GGG GAG AGA AGA ACA −3' which flank the putative CREB3 binding site and result in a 472 bp amplification product. qPCR was carried out on a QuantStudio 7 Flex (Applied Biosystems) using SYBR green (Abclonal, RK21203).

## Detection of fatty acylation on protein of interest using Alk14

HEK293T cells were transfected with IcsB and the gene of interest for 24 h, and then treated with Alk14 for 6 h before harvest. Alternatively, HEK293T cells transfected with gene of interest overnight were infected with M90T in the presence of Alk14 for 3 h. The cell pellets were processed as previously described[18].

## Defatty-acylation assay by sirtuins in vitro

HEK293T cells were transfected with IcsB and substrate genes for 24 h and treated with Alk14 for 6 h before harvest. The immunoprecipitated substrate proteins with Alk14 labeling were pulled down using anti-Flag affinity gel. The affinity gel containing the pulled down proteins was suspended in 25 µl of assay buffer (50 mM Tris-HCl, pH 8.0, 100 mM NaCl, 2 mM MgCl$_2$, 1 mM DTT) with 5 µM of SIRT2, with or without 1 mM NAD$^+$. The defatty-acylation reaction was allowed to proceed for 30 min at 37 °C and quenched by washing the affinity gel with IP wash buffer (25 mM Tris- HCl pH 8.0, 150 mM NaCl, 0.2% Nonidet P-40).

For click chemistry and in gel fluorescence, the immunopurified proteins with Alk14 labeling was re-suspended in 20 µL IP washing buffer. Rh-N3 (3 µL of 1 mM solution in DMF, final concentration 200 µM) was added to the above suspension, followed by the addition of TBTA (1 µL 10 mM solution in DMF, final concentration 500 µM), CuSO$_4$ (1 µL of 40 mM solution in H$_2$O, final concentration 2 mM), and TCEP (1 µL of 40 mM solution in H$_2$O, final concentration 2 mM). The click chemistry was allowed to proceed at room temperature for 30 min. The reaction mixture was mixed with 10 µL of 6 × protein loading buffer and heated at 95 °C for 10 min. After centrifugation at 16,000 g for 2 min at room temperature, 15 µL of the supernatant was treated with hydroxylamine (pH 7.4, 1 µL of 5 M solution in H$_2$O, final concentration 300 mM) or equivalent volume of water (negative control) at 95 °C for 7 min. The samples were resolved by SDS–PAGE. Rhodamine fluorescence signal was recorded by Typhoon 9400 Variable Mode Imager (GE Healthcare Life Sciences, Piscataway, NJ) with setting of Green (532 nm)/580BP30 PMT 500 V (normal sensitivity), and analyzed by Fiji software. Quantity One (Bio-Rad, Hercules, CA) was used for quantification of the fluorescence intensity.

SIRT2 was expressed and purified from *E. coli*. Human SIRT2 (aa38-356) was cloned and inserted into pET28a vector for the expression of N-terminal His$_6$-SUMO fusion protein. The SIRT2 protein was expressed in *E. coli* BL21 and purified using Ni affinity column purification. The desired fractions were pooled, concentrated and buffer exchanged, and the His6-SUMO tag was removed by overnight incubation at 4 °C with ULP1, followed by Ni-affinity column purification to remove any undigested SIRT2. The tag-free SIRT2 was further purified on a Superdex 75 column (Bio-Rad, Hercules, CA). The protein was eluted with 20 mM Tris-HCl, pH 8.0, 500 mM NaCl. After concentration, the target protein was frozen and stored at −80 °C.

## GDI pulldown

GST-RhoGDI were purified from *E. coli* strain BL21(DE3) cells using glutathione resin. For the pulldown assay, 500 µg lysates from transfected cells were incubated with 60 µg GST-RhoGDI pre-coupled to glutathione resin. After three washes in IP wash buffer, the bound proteins were eluted and analyzed by immunoblotting.

## Immunofluorescence

MEF cells were infected with M90T at MOI of 100:1 in glass bottom dishes (MatTek, Ashland MA). After 3 h, cells were washed with PBS three times and fixed with pre-chilled methanol for 10 min. Then the cells were permeabilized with 0.25% Triton X-100 for 10 min, blocked with 1% (w/v) BSA/0.1% Tween-20 in PBS for 1 h and incubated with LC3 antibody overnight. The cells were washed with 0.1% Tween-20 in PBS three times and incubated with Cy3-conjugated goat anti-rabbit IgG (H + L) secondary antibody for 1 h. The cells were washed with 0.1% Tween-20 in PBS three times and mounted with Fluoromount-G (SouthernBiotech, Homewood AL) containing DAPI before imaged on Cytation 5 (BioTek, Winooski VT). The number of MEF cells that contained *Shigella* and the number of MEF cells that contained LC3-postive *Shigella* were counted manually.

## Intraperitoneal shigellosis mice model study

The *S. flexneri* intraperitoneal infection mouse study was done as previously described[50]. Briefly, 6-week old wildtype and SIRT2 knockout C57/B6J mice (Jackson Lab, *Sirt2*$^{tm1.1Fwa}$) were intraperitoneally administrated with 150 *million Shigella* per 20 g of mouse body weight. After 17 h, mice were sacrificed, and tissue were harvested. The peritoneal wash was also collected. Tissues were washed in PBS with gentamycin (50 µg/ml) to remove superficial bacteria, rinsed twice in antibiotic-free PBS, and then mechanically homogenized in PBS with 2.5% Triton X-100 (200 mg of tissue per ml of PBS). The tissue extracts were diluted and plated onto Hektoen Enteric Agar plate. Tissue extracts and peritoneal wash from mice without *Shigella* infection were used as blank control. Colonies were counted after 18 h of culture at 37 °C. All animal experiments were approved by Cornell University's Institutional Animal Care and Use Committee.

## Intranasal shigellosis mice model study

The *S. flexneri* intranasal infection mouse study was done as previously described[13]. Briefly, 6–8-week-old wildtype and SIRT2 knockout C57/B6J mice were intranasally administrated with 1 *million Shigella*. After 3 days (or at indicated time points), mice were sacrificed, and lung tissue were harvested. The bronchoalveolar lavage (BAL) fluid was also collected. Tissues were mechanically homogenized in PBS with 2.5% Triton X-100. The tissue extracts were diluted and plated onto Hektoen Enteric Agar plate. Tissue extracts and BAL fluid from mice without *Shigella* infection were used as blank control. Colonies were counted after 18 h of culture at 37 °C. All animal experiments were approved by Cornell University's Institutional Animal Care and Use Committee.

## Statistics and reproducibility

Statistical analysis and generation of graphs was done with GraphPad Prism 9. *p*-values are shown on relevant graphs to indicates the significance of experimental results as calculated by the methods indicated in the figure legends. ***$p$-values < 0.001, below the limit of presentation by the statistics software. Shown blots and gel images are representative of at least three biological replicates.

## Reporting summary

Further information on research design is available in the Nature Research Reporting Summary linked to this article.

## Data availability

The data that support this study are available from the corresponding author upon reasonable request. Source data are provided with this paper.

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

## Acknowledgements
We thank Prof. Neal M. Alto for providing *S. flexneri* M90T strains and Dr. Toren Finkel for providing the *SIRT2*$^{+/+}$ and *SIRT2*$^{-/-}$ MEF cells. pKD46 and pKD4 plasmids are kind gifts from Dr. Tobias Doerr. This work is supported in part by an NIH/NIDDK grant DK107868.

## Author contributions
M.W. designed and performed experiments except those noted below. Y.Z. and G.P.K. performed a variety of experiments required for the revision. J.C. made the initial discovery that SIRT2 is able to remove RID mediated lysine fatty acylation. G.P.K., X.L., M.Z., and Y.Z. performed the mice study. T.Y. and D.H. contributed to studies involving BMDM. N.A.S. cloned multiple mammalian expression vectors encoding SIRT2 substrates. M.Y. synthesized Alk14. I.R.P. purified SIRT2 recombinant protein. H.L. directed the studies. M.W. and H.L. wrote the manuscript with input from all authors. All authors reviewed and approved the manuscripts.

## Competing interests
HL is a founder and consultant of Sedec Therapeutics. Other authors have no competing interests.
