## [Peer Review File · Nature Communications]

REVIEWER COMMENTS

Reviewer #1 (Remarks to the Author):

The manuscript by Wang M et al describes the analysis of SIRT2 deacylation activity in the context of Shigella infection. Their analysis of Golgi stress-induced expression of sirloins revealed that SIRT2 is specifically up-regulated via its CREB3 binding site. This led the authors to hypothesize that Shigella infection and its secreted virulence factors may up-regulate and be antagonized by SIRT2. Indeed, the authors show expression of Shigella IpaJ demyristoyl protease up-regulated SIRT2 expression. The authors then show that the acyltransferase activity of Shigella IcsB is antagonized by the SIRT2 deacylase activity for multiple substrates and impacts the protein-protein interactions between Rho GTPases and Rho GDI, which is associated with suppression of Shigella autophagosome escape. Finally, the authors demonstrate that SIRT2-KD and KO cells and mice are more susceptible to Shigella infection, which is abrogated in IcsB mutant Shigella. Overall, this study is well-executed and presented with appropriate methods and statistical analysis. Although the effects of SIRT2 and IcsB phenotypes are modest overall, the authors multiple lines (biochemical, cellular and in vivo analyses) of investigation provide a convincing argument for their findings. I am pleased to recommend publication in Nat Comm with minor changes to the figures highlighted below.

In Figures 6 - 8, the authors should correct the labels of Shigella mutant strains and convert "d" to "open triangle".

Reviewer #2 (Remarks to the Author):

In the process of exploring the regulation of SIRT2, an NAD⁺-dependent protein lysine deacylase, Wang et al. discovered that induction of Golgi stress resulted in the upregulation of SIRT2, likely through the transcription factor CREB. Intracellular pathogens such as Shigella flexneri, induce Golgi fragmentation during infection and the authors postulated a role for SIRT2 in the host defense response against Shigella. Shigella translocates effector proteins via a Type 3 secretion system (T3SS) into the host cell to disrupt host cell signaling and promote bacterial survival. IpaJ induces Golgi stress via proteolytic cleavage of glycine myristoylated ARFs and IcsB is a fatty acyl transferase that modifies several proteins during Shigella infection. Ectopic expression of IpaJ induced Golgi stress and increased SIRT2 levels. Moreover, infection of cells and mice with Shigella increased SIRT2 levels, which appeared to be dependent on IpaJ. Convincing in vitro and cell data suggests that SIRT2 can counter the activity of IcsB

by removing fatty acylation from several IcsB substrates. Finally, the authors show that SIRT2 can limit the escape of Shigella from autophagosomes and that SIRT2 limits Shigella infection in cells and in mice by counteracting IcsB.

Overall, this is a very nice study that identifies SIRT2 as an antibacterial host protein that appears to counteract Shigella infection. I have a few points the authors should consider when preparing a revised manuscript.

1. Do levels of protein fatty acylation increase in SIRT2 KO cells infected with Shigella and if so, do levels return to normal in host cells infected with IcsB KO Shigella?

The data in Supp fig 4 should be moved to the main figs and described as a modest increase in fatty acylation in IpaJ null.

2. There are several bacterial effector proteins that induce Golgi fragmentation from diverse pathogens. Can the authors demonstrate that Golgi fragmentation using an effector from a different pathogen also induces SIRT2 expression? This would broaden the scope of the paper and may prompt searches in other pathogens for fatty acyl transferase effectors.

3. Fig. 3A Does IpaJ expression in CREB knockdown cells increase SIRT2 expression?

4. Are CREB knockdown cells more susceptible to Shigella infection? If so, could this be rescued by overexpression of WT SIRT2, but not an inactive mutant from a plasmid?

5. Fig 3B. It is hard to believe the quantitation on the right looking at the representative immunoblot. It appears as though there is a massive upregulation of SIRT2 in response to Shigella infection (Mock #4 could be a statistical outlier). Moreover, the decrease in SIRT2 in IpaJ null Shigella is not very convincing. Surely, there are other T3SS effectors in Shigella that induce Golgi fragmentation/stress that could signal to increase SIRT2 expression. This should be addressed in the revised manuscript.

6. The data in Figure 8 is convincing; however, it is important to rescue these phenotypes by reintroducing IcsB (and the inactive mutant) into the icsB knockout strain to rule out any polar effects.

Reviewer #3 (Remarks to the Author):

The authors report a protective role for SIRT2 lysine defatty-acylase activity during Shigella infection. Golgi stress induced by infection, possibly through the virulence factor IpaJ, triggers CREB3-mediated increased expression of SIRT2. SIRT2 counterbalances IcsB-mediated fatty-acylation of a panel of target proteins including Rho GTPase and CHMP5, in turn limiting Shigella escape from autophagosome, and

improving host defenses. This observation was unexpected considering previous reports showing that genetic or pharmacologic inhibition of SIRT2 protected mice from listeriosis, salmonellosis, tuberculosis, and staphylococcal sepsis. Whether SIRT2 or its lysine defatty-acylase activity plays a distinctive during shigellosis remains open.

The basis of the study is that Golgi stress increases SIRT2 expression, which is nicely shown in vitro using drug- and Shigella-induced Golgi stress (Figures 1 and 3). A role for IpaJ in that process seems obvious in the infectious setting. On the contrary, in vivo data are not very convincing (Figure 3B and supplementary Figure 2). Western blots show similar levels of SIRT2 in BAL cells from mice challenged i.n. with IpaJ deficient and wild-type Shigella (Figure 3B, left panel). The lower upregulation of SIRT2 with IpaJ deficient Shigella depicted in bar graph results from an increased expression of HSP90 used for normalization. More annoyingly, there is barely any upregulation of SIRT2 in colonic tissues from mice challenged i.p. with Shigella, a model developed to mimic human bacillary dysentery (reference #50). It would be worthwhile to test other routes of infection (orally or i.g.) that better mimic natural infection. Unless more robust in vivo data are presented, the authors need to revise their conclusions concerning the impact of Shigella and IpaJ on SIRT2 expression during shigellosis.

SIRT2 is a powerful deacetylase. This raises question whether the effects reported here are related to one or the other of the enzymatic activities of SIRT2 when using SIRT2 deficient cells or mice. To address this issue, the authors could compare the effects of drugs targeting deacetylase activity (AGK2, TM, NPD11033), defatty-acylase activity (S2DMi-6, -7, -9) and both deacetylase and defatty-acylase activities (TM-P4-Thal) of SIRT2.

SIRT6 and HDAC11 also exhibit a lysine defatty-acylase activity. Are these enzymes modulated during Shigella infection in vivo? Could SIRT6 or HDAC11 play a role during Shigella infection?

Bacteria such as enteropathogenic Escherichia coli, Chlamydia, Rickettsia or Group A Streptococcus cause Golgi fragmentation and inhibit anterograde transport in infected host cells. Do they also increase SIRT2 (possibly SIRT6 and HDAC11) expression upon infection?

Abstract: Remove "Foreign" (first word).

Methods:

Some key steps of the gentamycin assay are missing. Which MOI was used? Using how many cells? Without this information, we cannot interpret the results shown in Figure 7. I could not find the description of the generation of CREB3 N-ter and SIRT2 promoter luciferase reporter constructs. A more "relevant" cell line such as Caco-2/TC7 cells could be tested in some experiments, for example looking at SIRT2 and CREB3 expression.

References 13 and 51 are identical (Burnaevskiy et al., Nature 2013).

Figure 1C and supplementary material. Please provide data supporting your statement that (golgicide A) Exo1 increases SIRT2 protein level.

Figure 1C. Mention which cell line was used.

Figure 2B. Show that restoration of CREB3 using the CREB3 expression construct increases SIRT2 protein levels in CREB3 KD A549 cells.

Figure 2D. Evidence over expression of CREB3 N-ter by western blotting. The role of CREB3 binding site and CREB3 should be strengthened by showing: 1) that a CREB3 mutant SIRT2 promoter is less transcriptionally active, and 2) ChIP assays indicating CREB3 binding to the SIRT2 promoter region.

Figure 7. Panel A: The SEM for SIRT2^{+/+} at 200 min is strange.

Figure 7. Panels B and C: Was it one experiment? Please specify in the legend.

Figure 7. Panel E: Which time point? Show cytokine levels in control mice, i.e. mice challenged with PBS or NaCl. Did you measure other cytokines, especially IL-1 β ?

Figure 7. Reverse panels E and F to fit the flow of your narration.

Figure 7 and 8. Replace MOI by the actual number of bacteria recovered, i.e. the CFUs.

We would like to thank the reviewers for their careful and helpful comments. As detailed in the point-by-point response below, we have addressed the comments and believe that the manuscript is now more rigorous after addressing the comments. The reviewers' comments are shown in black fonts while our response are shown in blue.

Reviewer 1

Comment: The manuscript by Wang M et al describes the analysis of SIRT2 deacylation activity in the context of Shigella infection. Their analysis of Golgi stress-induced expression of sirloins revealed that SIRT2 is specifically up-regulated via its CREB3 binding site. This led the authors to hypothesize that Shigella infection and its secreted virulence factors may up-regulate and be antagonized by SIRT2. Indeed, the authors show expression of Shigella IpaJ demyristoyl protease up-regulated SIRT2 expression. The authors then show that the acyltransferase activity of Shigella IcsB is antagonized by the SIRT2 de-acylase activity for multiple substrates and impacts the protein-protein interactions between Rho GTPases and Rho GDI, which is associated with suppression of Shigella autophagosome escape. Finally, the authors demonstrate that SIRT2-KD and KO cells and mice are more susceptible to Shigella infection, which is abrogated in IcsB mutant Shigella. Overall, this study is well-executed and presented with appropriate methods and statistical analysis. Although the effects of SIRT2 and IcsB phenotypes are modest overall, the authors multiple lines (biochemical, cellular and in vivo analyses) of investigation provide a convincing argument for their findings. I am pleased to recommend publication in Nat Comm with minor changes to the figures highlighted below.

In Figures 6 - 8, the authors should correct the labels of Shigella mutant strains and convert "d" to "open triangle".

Response: "d" was replaced with "Δ" in relevant graphs. We thank the reviewer's careful suggestions in fixing figures in the manuscript.

Reviewer 2

In the process of exploring the regulation of SIRT2, an NAD⁺-dependent protein lysine deacylase, Wang et al. discovered that induction of Golgi stress resulted in the upregulation of SIRT2, likely through the transcription factor CREB. Intracellular pathogens such as Shigella flexneri, induce Golgi fragmentation during infection and the authors postulated a role for SIRT2 in the host defense response against Shigella. Shigella translocates effector proteins via a Type 3 secretion system (T3SS) into the host cell to disrupt host cell signaling and promote bacterial survival. IpaJ induces Golgi stress via proteolytic cleavage of glycine myristoylated ARFs and IcsB is a fatty acyl transferase that modifies several proteins during Shigella infection. Ectopic expression of IpaJ induced Golgi stress and increased SIRT2 levels. Moreover, infection of cells and mice with Shigella increased SIRT2 levels, which appeared to be dependent on IpaJ. Convincing in vitro and cell data suggests that SIRT2 can counter the activity of IcsB by removing fatty acylation from several IcsB substrates. Finally, the authors show that SIRT2 can limit the escape of Shigella from autophagosomes and that SIRT2 limits Shigella infection in cells and in mice by counteracting IcsB.

Overall, this is a very nice study that identifies SIRT2 as an antibacterial host protein that appears to counteract Shigella infection. I have a few points the authors should consider when preparing a revised manuscript.

Comment 1: Do levels of protein fatty acylation increase in SIRT2 KO cells infected with Shigella and if so, do levels return to normal in host cells infected with IcsB KO Shigella? The data in Supp fig 4 should be moved to the main figs and described as a modest increase in fatty acylation in IpaJ null.

Response: To address this comment, we infected either control or SIRT2 KD HEK 293T cells with WT or IcsB KO Shigella and examined total fatty acylation by global Alk14 labeling. WT Shigella infection increased the fatty acylation (especially around 20 kDa where the small GTPases reside) and SIRT2 KD further increased the fatty acylation. The increase in fatty acylation was not observed when infected with IcsB KO. The data was included in the manuscript as Supp Fig 4C.

The data in the original Supp Fig 4 was moved to the main Fig 4D-E and the text has been changed to reflect this.

Comment 2: There are several bacterial effector proteins that induce Golgi fragmentation from diverse pathogens. Can the authors demonstrate that Golgi fragmentation using an effector from a different pathogen also induces SIRT2 expression? This would broaden the scope of the paper and may prompt searches in other pathogens for fatty acyl transferase effectors.

Response: Enteropathogenic *E. coli* toxin EspG and *S. flexneri* toxin VirA are both known to cause golgi fragmentation. When overexpressed in HEK 293T cells, both increased the level of SIRT2 protein. This data is included in Supp Fig 2G.

Comment 3: Fig. 3A Does IpaJ expression in CREB knockdown cells increase SIRT2 expression?

Response: IpaJ was transfected into control and ShCREB3 A549 cells. Control cells demonstrated a marked increase in SIRT2 levels whereas ShCREB3 cells did not. This data is included in Supp Fig 2E.

Comment 4: Are CREB knockdown cells more susceptible to Shigella infection? If so, could this be rescued by overexpression of WT SIRT2, but not an inactive mutant from a plasmid?

Response: Infection of ShCREB3 A549 cells resulted in a greater number of bacteria in a gentamicin protection assay. Overexpression of WT SIRT2 in ShCREB3 cells partially rescued this effect while H187Y SIRT2 does not. This data is included in Supp Fig 6D.

Comment 5: Fig 3B. It is hard to believe the quantitation on the right looking at the representative immunoblot. It appears as though there is a massive upregulation of SIRT2 in response to Shigella infection (Mock #4 could be a statistical outlier). Moreover, the decrease in SIRT2 in IpaJ null Shigella is not very convincing. Surely, there are other T3SS effectors in Shigella that induce

Golgi fragmentation/stress that could signal to increase SIRT2 expression. This should be addressed in the revised manuscript.

Response: We agree with the reviewers that the mouse data has larger variations due to various reasons. To address this comment, we infected bone marrow-derived macrophages (BMDMs) from WT C57BL/6 mice. We believe this is a cleaner experiment compared to infecting mice and isolating tissue samples from infected mice. Tissue samples contain a variety of different cell types that may respond differently to infection. BMDMs infected with WT *Shigella* showed increased levels of SIRT2 compared to Δ ipaj or Δ ipaj Δ vira *Shigella*. This data is included in Fig 3B and Supp Fig 2B.

Comment 6: The data in Figure 8 is convincing; however, it is important to rescue these phenotypes by reintroducing IcsB (and the inactive mutant) into the icsB knockout strain to rule out any polar effects.

Response: WT or C306A IcsB was reintroduced to IcsB KO M90T bacteria. SIRT2 KO mice had much more bacteria than WT mice in lung samples following intranasal infection of M90T with WT IcsB. SIRT2 KO mice had no difference in bacteria from WT mice when infected with M90T with C306A IcsB. This data is included in Fig 8C.

Reviewer 3

The authors report a protective role for SIRT2 lysine defatty-acylase activity during *Shigella* infection. Golgi stress induced by infection, possibly through the virulence factor IpaJ, triggers CREB3-mediated increased expression of SIRT2. SIRT2 counterbalances IcsB-mediated fatty-acylation of a panel of target proteins including Rho GTPase and CHMP5, in turn limiting *Shigella* escape from autophagosome, and improving host defenses. This observation was unexpected considering previous reports showing that genetic or pharmacologic inhibition of SIRT2 protected mice from listeriosis, salmonellosis, tuberculosis, and staphylococcal sepsis. Whether SIRT2 or its lysine defatty-acylase activity plays a distinctive during shigellosis remains open.

Comment 1: The basis of the study is that Golgi stress increases SIRT2 expression, which is nicely shown in vitro using drug- and *Shigella*-induced Golgi stress (Figures 1 and 3). A role for IpaJ in that process seems obvious in the infectious setting. On the contrary, in vivo data are not very convincing (Figure 3B and supplementary Figure 2). Western blots show similar levels of SIRT2 in BAL cells from mice challenged i.n. with IpaJ deficient and wild-type *Shigella* (Figure 3B, left panel). The lower upregulation of SIRT2 with IpaJ deficient *Shigella* depicted in bar graph results from an increased expression of HSP90 used for normalization. More annoyingly, there is barely any upregulation of SIRT2 in colonic tissues from mice challenged i.p. with *Shigella*, a model developed to mimic human bacillary dysentery (reference #50). It would be worthwhile to test other routes of infection (orally or i.g.) that better mimic natural infection. Unless more robust in vivo data are presented, the authors need to revise their conclusions concerning the impact of *Shigella* and IpaJ on SIRT2 expression during shigellosis.

Response: While we agree with the reviewer that the mouse data in the original Figure 3B (now supplementary Figure 2B) has some variation due to the complexity of the mouse experiments (for example, the amount of the BAL cells collected from each mouse are limited and prevent us

from adjusting the loading to get better blots), we emphasize that the overall quantification (which normalizes SIRT2 levels by HSP90 levels) and statistics support our conclusion that SIRT2 is upregulated by *Shigella* infection and the IpaJ deletion strain induced less SIRT2.

To further address this comment, we infected BMDMs obtained from WT C57BL/6 mice and infected the BMDMs. We believe this is a cleaner experiment as isolating tissue samples from infected mice. Tissue samples contain a variety of different cell types that may be responding differently to infection. BMDMs infected with WT *Shigella* showed increased levels of SIRT2 compared to Δ ipaj or Δ ipaj Δ vira *Shigella*. This data is inserted into Fig 3B and Supp Fig 2B.

Comment 2: SIRT2 is a powerful deacetylase. This raises question whether the effects reported here are related to one or the other of the enzymatic activities of SIRT2 when using SIRT2 deficient cells or mice. To address this issue, the authors could compare the effects of drugs targeting deacetylase activity (AGK2, TM, NPD11033), defatty-acylase activity (S2DMI-6, -7, -9) and both deacetylase and defatty-acylase activities (TM-P4-Thal) of SIRT2.

Response: We thank the reviewer for the interesting suggestion. However, due to a recent finding, we think that this experiment would not provide the information to address this question. It is true that our lab and others reported that several SIRT2 inhibitors can only affect deacetylation (such as TM) while other inhibitors can affect both deacetylation and demyristoylation. These conclusions were based on limited substrates in vitro and in cells. We recently found that the reality is more complicated as we found that TM, which previously thought to only inhibit deacetylation, can also effectively inhibit the demyristoylation of ARF6 by SIRT2 (*Nat. Comm.*, 11:1067, 2020). Thus, the specificity of the inhibitors is not only determined by the acyl group, but also the substrate protein. In other words, we cannot assume that AGK2/TM/NPD11033 do not inhibit the deacylation of the proteins that are acylated by IcsB and deacylated by SIRT2.

However, we agree with the reviewer that it is possible that the deacetylation activity of SIRT2 could also be important. We have included a sentence in the discussion to state that although this work focuses on the defatty-acylation activity of SIRT2, it is possible that the deacetylation activity of SIRT2 is also important for fighting *Shigella* infection.

Comment 3: SIRT6 and HDAC11 also exhibit a lysine defatty-acylase activity. Are these enzymes modulated during *Shigella* infection in vivo? Could SIRT6 or HDAC11 play a role during *Shigella* infection? Bacteria such as enteropathogenic *Escherichia coli*, *Chlamydia*, *Rickettsia* or Group A *Streptococcus* cause Golgi fragmentation and inhibit anterograde transport in infected host cells. Do they also increase SIRT2 (possibly SIRT6 and HDAC11) expression upon infection?

Response: SIRT6 is not able to reduce IcsB-catalyzed fatty acylation on CHMP5. HDAC11 can reduce IcsB-catalyzed fatty acylation on CHMP5 (data included in Supp Fig 3E). HDAC11 may thus affect *Shigella* infection, but it is not upregulated by Golgi stress (Fig 1G) and thus falls outside the scope of this particular manuscript. Enteropathogenic *E. coli* toxin EspG and *S. flexneri* toxin VirA are both known to cause Golgi fragmentation. When overexpressed in HEK 293T cells, both increased the level of SIRT2 protein. Thus, it is possible that SIRT2 upregulation is a shared mechanism during several bacterial infections. This data is included in Supp Fig 2G. Moreover, Infection of A549 cells with enteropathogenic *E. coli* upregulated SIRT2, consistent with a conserved mechanism spanning multiple pathogenic bacteria. This data is included in Supp Fig 2F.

Comment 4:

Abstract: Remove “Foreign” (first word).

Methods:

Some key steps of the gentamycin assay are missing. Which MOI was used? Using how many cells? Without this information, we cannot interpret the results shown in Figure 7. I could not find the description of the generation of CREB3 N-ter and SIRT2 promoter luciferase reporter constructs. A more “relevant” cell line such as Caco-2/TC7 cells could be tested in some experiments, for example looking at SIRT2 and CREB3 expression.

References 13 and 51 are identical (Burnaevskiy et al., Nature 2013).

Response: We thank the reviewer for the helpful comments and revised the manuscript accordingly.

Figure 1C and supplementary material. Please provide data supporting your statement that (golgicide A) Exo1 increases SIRT2 protein level.

Response: The statement concerning Exo1 was removed as it does not add anything to the manuscript.

Figure 1C. Mention which cell line was used.

Response: We thank the reviewer for the helpful comments and added the cell lines used.

Figure 2B. Show that restoration of CREB3 using the CREB3 expression construct increases SIRT2 protein levels in CREB3 KD A549 cells.

Response: Data demonstrating this point was collected and inserted to the manuscript as Fig 2C.

Figure 2D. Evidence over expression of CREB3 N-ter by western blotting. The role of CREB3 binding site and CREB3 should be strengthened by showing: 1) that a CREB3 mutant SIRT2 promoter is less transcriptionally active, and 2) ChIP assays indicating CREB3 binding to the SIRT2 promoter region.

Response: A blot of CREB3 (1-220) is added into Fig 2E. To address the transcriptional activity of the SIRT2 promoter, we carried out a luciferase assay using a control pGL3-994 vector or one with the 5' SIRT2 promoter region inserted. The SIRT2 promoter construct was more transcriptionally active and displayed enhanced signal in response to BFA treatment. The empty vector showed no change with BFA treatment. The data is shown in Fig 2F. To demonstrate CREB3 binding to the SIRT2 promoter, we designed primers to flank the putative CREB3 binding site and carried out ChIP using a CREB3 antibody with and without BFA treatment. qPCR demonstrated more enrichment of the SIRT2 promoter in ChIP samples from BFA treated cells. The data is included in Fig 2G.

Figure 7. Panel A: The SEM for SIRT2+/+ at 200 min is strange.

Response: There was a user error when plotting the original data. The graph has been updated to reflect the actual data. We thank the reviewer for the careful observation.

Figure 7. Panels B and C: Was it one experiment? Please specify in the legend.

Response: The data was collected from multiple experiments and is now indicated in the legend. The SEM for these experiments was remarkably low and is not apparent on the graph, which caused the confusion.

Figure 7. Panel E: Which time point? Show cytokine levels in control mice, i.e. mice challenged with PBS or NaCl. Did you measure other cytokines, especially IL-1 β ?

Response: The collection time (two days) was added to the figure legend. Cytokine levels from mice challenged with PBS fell below the limit of detection for the experimental setup. This does change affect out observation of lower cytokine levels in SIRT2 KO mice, but this information has been added to the figure legend. Other cytokines were not measured. We did not measure IL-1 β .

Figure 7. Reverse panels E and F to fit the flow of your narration.

Response: Changed as suggested.

Figure 7 and 8. Replace MOI by the actual number of bacteria recovered, i.e. the CFUs.

Response: Changed as suggested.

REVIEWERS' COMMENTS

Reviewer #3 (Remarks to the Author):

The authors have done a great job addressing my concerns. I recommend publication.

Reviewer #4 (Remarks to the Author):

The authors addressed my comments and improved their manuscript. Congratulations. One comment however has been eluded, i.e. about the route of infection used in the preclinical model. While informative for deciphering local immune protection against *Shigella* infection, the mouse lung model does not mimic bacillary dysentery, which is the usual disease caused by *Shigella* infection in humans. This should be mentioned as a limitation of the study in the discussion.

We thank all the reviewers for their positive comments on the manuscript. All reviewers have recommended publication. Reviewer 4 has one final comment: “One comment however has been eluded, i.e. about the route of infection used in the preclinical model. While informative for deciphering local immune protection against *Shigella* infection, the mouse lung model does not mimic bacillary dysentery, which is the usual disease caused by *Shigella* infection in humans. This should be mentioned as a limitation of the study in the discussion.”

We agree with this comments and have added the following sentence in the discussion:

“One limitation of our study is that we used a mouse lung infection model for *Shigella*. While this model is useful for deciphering local immune protection, it does not mimic bacillary dysentery, the usual disease caused by *Shigella* infection in humans.”

We have also updated the author checklist and reporting summary. Please let us know if anything else is needed from us.